# In Vitro Antimicrobial Activity of Lavender, Mint, and Rosemary Essential Oils and the Effect of Their Vapours on Growth of *Penicillium* spp. in a Bread Model System

**DOI:** 10.3390/molecules26133859

**Published:** 2021-06-24

**Authors:** Veronika Valková, Hana Ďúranová, Lucia Galovičová, Nenad L. Vukovic, Milena Vukic, Miroslava Kačániová

**Affiliations:** 1AgroBioTech Research Centre, The Slovak University of Agriculture in Nitra, Tr. A. Hlinku 2, 94976 Nitra, Slovakia; hana.duranova@uniag.sk; 2Department of Fruit Sciences, Viticulture and Enology, Faculty of Horticulture and Landscape Engineering, Slovak University of Agriculture, Tr. A. Hlinku 2, 94976 Nitra, Slovakia; l.galovicova95@gmail.com; 3Department of Chemistry, Faculty of Science, University of Kragujevac, P.O. Box 12, 34000 Kragujevac, Serbia; nvukovic@kg.ac.rs (N.L.V.); milena.vukic@pmf.kg.ac.rs (M.V.); 4Department of Bioenergy, Food Technology and Microbiology, Institute of Food Technology and Nutrition, University of Rzeszow, Zelwerowicza St. 4, 35601 Rzeszow, Poland

**Keywords:** essential oils, volatile compounds, antioxidant activity, antifungal activity, antibacterial activity, bakery product, moisture content, water activity

## Abstract

The chemical composition, antioxidant activity, and antimicrobial properties of three commercially available essential oils: rosemary (REO), lavender (LEO), and mint (MEO), were determined in the current study. Our data revealed that the major components of REO, MEO, and LEO were 1,8-cineole (40.4%), menthol (40.1%), and linalool acetate (35.0%), respectively. The highest DPPH radical-scavenging activity was identified in MEO (36.85 ± 0.49%) among the investigated EOs. Regarding antimicrobial activities, we found that LEO had the strongest inhibitory efficiencies against the growth of *Pseudomonas aeruginosa* and *Candida* (*C.*) *tropicalis*, MEO against *Salmonella* (*S.*) *enterica*, and REO against *Staphylococcus* (*S.*) *aureus*. The strongest antifungal activity was displayed by mint EO, which totally inhibited the growth of *Penicillium* (*P.*) *expansum* and *P. crustosum* in all concentrations; the growth of *P. citrinum* was completely suppressed only by the lowest MEO concentration. The lowest minimal inhibitory concentrations (MICs) against *S. enterica*, *S. aureus*, and *C. krusei* were assessed for MEO. In situ analysis on the bread model showed that 125 µL/L of REO exhibited the lowest mycelial growth inhibition (MGI) of *P. citrinum*, and 500 µL/L of MEO caused the highest MGI of *P. crustosum*. Our results allow us to make conclusion that the analysed EOs have promising potential for use as innovative agents in the storage of bakery products in order to extend their shelf-life.

## 1. Introduction

Bread is an important staple food worldwide. However, its fungal spoilage during storage is a serious problem that can result not only in economic losses, but also in human health hazards because of the presence of mycotoxins [1]. In general, bread rot is caused by microscopic fungi, such as *Penicillium* and *Aspergillus*, as well as *Mucor*, *Cladosporium*, *Fusarium*, and *Rhizopus* [2]. One of the potential alternatives to prevent the spoilage of bakery goods appears to be the application of essential oils (EOs) as natural preservatives [3].

EOs are volatile secondary metabolites derived from plants responsible for their typical smell and taste. They can be obtained from about 17,500 angiosperm plants (e.g., Rutaceae, Lamiaceae, Zingiberaceae, Myrtaceae, Asteraceae), and among them, only approximately 300 species of EOs are commercially available [4]. These highly concentrated aromatic materials can be extracted from various parts of the plant, including the leaves, stem, flowers, seeds, roots, fruit rind, resin, or bark [5]. The isolation of such oils is relatively simple, but their chemical composition depends on the extraction technique used. Hydrodistillation, solvent extraction, simultaneous distillation-extraction, supercritical carbon dioxide extraction, and the use of microwave ovens are the most frequently used extraction methods [6].

There are some properties, such as radical scavenging [7], as well as antiviral [8], antiprotozoal [9], antibacterial [10], and antifungal [11] properties, and many others that are well recognized regarding the biological activities of EOs. The wide range of activities is attributed to the diverse chemical composition of EOs. Generally, lipophilic and highly volatile components from many chemical classes are the most common substances found in EOs [12]. Terpenic and phenolic compounds [13], as well as alcohols and esters have shown significant biological effects [14].

Rosemary (*Rosmarinus officinalis* L.; Lamiaceae) is a rich source of phenolic compounds, such as carnosol, rosmanol, rosmaridiphenol, and rosmarquinone [15]. It is used in the treatment of various disorders and in food preservation as well [16]. Rosemary EO (REO) possesses a higher antioxidant activity than other EOs [15].

Lavender (*Lavandula* spp.) is a herb that belongs to the family Labiatae and is intensively cultivated for oil production [17]. The products derived from this popular garden herb are used as therapeutics as well as antibacterial agents, and lavender essential oil (LEO) is traditionally believed to have sedative, carminative, antidepressant, and anti-inflammatory properties [18].

Mint (*Mentha piperita* L.; Lamiaceae) EO (MEO) has been used in traditional medicine and its biological activity may be due to its major volatile components: carvone, menthol, and menthone [19].

The major purpose of our study was to evaluate the antifungal effects of selected EOs (REO, LEO, and MEO) against *Penicillium* spp. using the contact vapour method. In addition, the volatile compounds of the EOs, their antimicrobial properties and antioxidant activities, and basic technological properties of bread (as a model substrate for growth of fungi in situ) were determined. Summarily, the EO with the greatest potential and its effective concentration applied as a natural preservative used in the storage of bread in commercial practice were assessed.

## 2. Results

The vapour-phase antifungal activities of three selected EOs obtained from rosemary, lavender, and mint against *Penicillium* spp. inoculated on bread samples were evaluated in the current study. The data expand our findings related to bread preservation using natural alternatives, such as EOs [20,21].

### 2.1. Chemical Composition of EOs

The chemical composition of our EOs was determined by gas chromatography/mass spectrometry (GC/MS) analysis. Overall, there were 44, 43, and 40 components accounting for a total of 99.4%, 99.6%, and 99.5% of the EO identified in the LEO, MEO, and REO, respectively (Table 1). The main components of the LEO were linalool acetate (35.0%), linalool (32.7%), and 1,8-cineole (8.1%). The major chemical constituents of the MEO were represented by the menthol (40.1%), menthone (16.8%), and menthyl acetate (9.1%). However, 1,8-cineole (40.4%), camphor (11.9%), and α-pinene (8.7%) were detected as major compounds in the REO.

### 2.2. Antioxidant Activity of EOs

Table 2 presents the quenching of 2,2-diphenyl-1-picrylhydrazyl (DPPH) radicals from which it is evident that all analysed EOs displayed moderate antioxidant activity. Additionally, the results indicate the significantly (*p* < 0.05) strongest DPPH-scavenging activity of mint EO (36.85 ± 0.49%), whilst the EOs from lavender and rosemary displayed lower values for the activity (29.08 ± 0.99%, 28.76 ± 2.68%, respectively) without statistically significant differences.

### 2.3. Antimicrobial Activity of EOs

A disc diffusion method was used to evaluate the antimicrobial activities of selected EOs (LEO, MEO, REO) against Gram-positive (G^+^) and Gram-negative (G^−^) bacteria, yeasts, and microscopic fungi in the current study. As shown in Table 3, our results revealed that LEO had the strongest inhibitory efficiency against the growth of *Pseudomonas* (*P.*) *aeruginosa* and *Candida* (*C.*) *tropicalis*, with a zone of inhibition of 9.33 ± 0.58 mm and 9.66 ± 0.58 mm, respectively, which were significantly (*p* < 0.05) higher than those of MEO (7.33 ± 1.53, 6.00 ± 0.00 mm, respectively) and REO (7.00 ± 1.00, 8.33 ± 0.58 mm, respectively). On the other hand, LEO exhibited the least antimicrobial activity against the remaining bacteria and yeasts, with an inhibition zone ranging from 1.00 ± 0.00 (*Yersinia* (*Y.*) *enterocolitica, Staphylococcus* (*S.*) *aureus*) to 6.33 ± 0.58 mm (*C. krusei*). The antimicrobial activities were statistically (*p* < 0.05) different in comparison with MEO and REO. The EO from *M. piperita* showed the strongest antimicrobial activity against *Salmonella* (*S.*) *enterica*, with an inhibition zone of 9.00 ± 1.00 mm, which was demonstrably (*p* < 0.05) higher than those exhibited by LEO and REO. The inhibitory actions of MEO against the growth of *Y. enterocolitica* (7.00 ± 1.00 mm), *Enterococcus* (*E.*) *faecium* (8.00 ± 1.00 mm), *C. glabrata* (7.33 ± 0.58 mm), *C. albicans* (6.67 ± 0.58 mm), and *C. krusei* (9.67 ± 0.58 mm) were similar to those of REO (7.67 ± 1.53, 8.00 ± 1.00, 7.67 ± 0.58, 8.00 ± 1.00, 10.00 ± 1.00 mm) but considerably (*p* < 0.05) higher as compared to the actions of LEO (1.00 ± 0.00, 3.00 ± 0.00, 2.00 ± 0.00, 2.00 ± 0.00, 6.33 ± 0.58 mm, respectively).

The strongest antimicrobial activity of REO was shown to be against *S. aureus*, with an inhibition zone of 10.33 ± 0.58 mm, which significantly differed from those of MEO and LEO. Additionally, the REO activities against *S. enterica* and *C. tropicalis* were considerably (*p* < 0.05) higher as compared to LEO and MEO, respectively.

Data from the inhibitory effects of the analysed EOs against three tested *Penicillium* (*P*.) spp. fungi (*P. crustosum*, *P. citrinum*, *P. expansum*) are shown in Table 4. Our results revealed that the growth inhibition of fungi strains depends on the type and concentration of the EO used. Remarkable antifungal activity was observed for the MEO among all investigated EOs, which completely inhibited the growth of *P. crustosum* and *P. expansum* in all used concentrations (125, 250, and 500 µL/L). The growth of *P. citrinum* was also totally inhibited by MEO in a concentration of 125 µL/L, whereas it showed significantly (*p* < 0.05) different zones of inhibition (6.67 ± 0.58 mm; 9.00 ± 1.00 mm, respectively) in the 250 and 500 µL/L concentrations. On the other hand, LEO (125 and 250 µL/L) and REO (in all concentrations) displayed no inhibitory impact on the growth of *P. crustosum*, and *P. citrinum* and *P. expansum* were significantly (*p* < 0.05) inhibited by the EOs in the highest concentrations.

### 2.4. Minimum Inhibitory Concentrations of EOs against Gram-Negative and Gram-Positive Bacteria, and Yeasts

The MIC values of tested EOs against Gram-negative and Gram-positive bacteria and yeasts are represented in Table 5 and Table 6. The EOs displayed a variable degree of inhibition activity against the different tested strains, with significant differences (*p* < 0.05) among the analysed EOs, as well as the microorganisms that were used. LEO had the lowest MICs against *C. albicans* regarding the effectiveness of selected EOs, whilst EO was the most effective against *S. enterica*. MEO exhibited the weakest action against the growth of *P. aeruginosa*, and also against *C. glabrata* and *C. albicans*. On the other hand, MEO was the most effective against *S. enterica* and *C. krusei*. Finally, REO had weak antimicrobial activity against *C. albicans* and a stronger effect against *S. enterica*.

### 2.5. Moisture Content and Water Activity of Bread Samples

The results from the moisture content (MC) and water activity (a_w_) measurements showed that the parameters of bread analysed in our study had values of 41.65 ± 0.55% and 0.944 ± 0.001, respectively.

### 2.6. In Situ Antifungal Analysis on Bread

The antifungal properties of the analysed EOs on bread are presented in Table 7 and Figure 1. The results of the analysis revealed that LEO had the significantly (*p* < 0.05) highest inhibition against *P. crustosum* in all used concentrations (125, 250, and 500 µL/L) in a very slightly increasing manner (81.18 ± 2.78%, 85.88 ± 1.95%, 88.64 ± 2.74%, respectively), and against *P. citrinum* and *P. expansum* in the highest concentrations (89.38 ± 2.05%, 86.12 ± 3.04%, respectively). The strongest significant (*p* < 0.05) antifungal activity of MEO was observed against *P. crustosum* in concentrations of 125 and 500 µL/L (87.91 ± 1.06% and 90.19 ± 2.99%, respectively), and against *P. citrinum* and *P. expansum* in the lowest and the highest concentrations, respectively. Interestingly, REO exhibited (*p* < 0.05) the highest activity in different concentrations for individual fungal species: *P. crustosum* in 250 µL/L (92.48 ± 1.69%), *P. citrinum* at 500 µL/L (57.36 ± 2.63%), and *P. expansum* in 125 µL/L (86.48 ± 3.55%).

## 3. Discussion

It is generally known that the antibacterial effects of EOs depend on their chemical composition [22], which can be influenced by various factors, such as the plant developmental state, the plant part used for extraction, plant geographical location, and physical and chemical characteristics of the soil and climate in question [23].

According to ISO [24], the EOs obtained from *L. angustifolia* are mainly composed of linalool acetate (25.0–47.0%) and linalool (20.0–45.0%), which is in line with our study (35.0% and 32.7%, respectively). Similarly, the research by Zheljazkov et al. [25] and Baydar and Kineci [26] showed that linalool (23.3–43.4%; 34.0%) and linalool acetate (20.2–39.6%; 47.7%) were the main components of the EO from *L. angustifolia* Mill. and *L. x intermedia Emeric ex Loisel*, respectively. Other components in lavender EOs [24] are present in smaller quantities, such as E-β-ocimene (0.0–10.0%), Z-β-ocimene (0.0–6.0%), 4-terpineol (0–8.0%), lavandulyl acetate (0.0–8.0%), 3-octanone (0.0–5.0%), lavandulol (0.0–3.0%), α-terpineol (0.0–2.0%), β-phellandrene (0.0–1.0%), and limonene (0.0–1.0%), which is also in agreement with our results (0.9%, 0.0%, 1.3%, 2.0%, 0.3%, 0.0%, 8.1%, 1.3%, 0.0%, 0.9%, respectively). On the other hand, the concentrations of 1,8-cineole (8.1%) and camphor (6.4%) were higher in our LEO as compared to those (0.0–3.0% and 0.0–1.5%, respectively) reported by ISO [24]. The higher content of both components in the EO can not only significantly affect its aroma intensity [27], but the higher abundance of 1,8-cineole can also be associated with stronger antifungal properties [28].

*Mentha* EOs consist mainly of oxygenated monoterpenes as a major fraction [29]. Soković et al. [30] determined in the EO from *Mentha piperita*, that the most abundant substances were menthol (37.4%), menthyl acetate (17.4%), and menthone (12.7%). The chemical composition of the EO from *M. piperita* during two seasons (summer and winter) was analysed in the study by Hussain et al. [29]. The authors found that the main components of the EOs collected during summer and winter were menthone (28.13% and 25.54%), menthyl acetate (9.51% and 9.68%), limonene (7.58% and 7.73%), and isomenthone (4.04% and 7.63%), respectively. However, the content of the compounds (menthone 16.8%, menthyl acetate 9.1%, menthol 40.1%, limonene 1.8%, and isomenthone 2.8%) was different in our study, indicating that the aforementioned factors might contribute to the discrepancies in the chemical composition of the MEO analysed in the three studies. We assume that a lower content of methyl acetate in our MEO as compared to that by Soković et al. [30] may be connected with its higher antifungal activity since it was found that this compound causes a decrease in the antifungal properties of the EOs [28].

Regarding the REO chemical composition, our findings are confirmed by the many other studies [31,32,33] in which 1,8-cineole, camphor, and α-pinene were reported to be the major components of *R. officinalis* EOs. However, the percentages of the individual components (43.99%, 26.54%, and 37.6% and 47.2% for 1,8-cineole; 12.41%, 12.88%, and 7.1% and 13.3% for camphor; 10.09%, 20.14%, and 7.0% and 19.4% for α-pinene) of the rosemary EOs were different compared to our REO (40.4%, 8.7%, and 11.9%, respectively). Similarly, Elamrani et al. [34] indicated that the major compounds of *R. eriocalix* oil were 1,8-cineole (54.6%), camphor (8.6%), and β-pinene (6.8%). The commercially known EO from *R. officinalis* is characterized by the presence of 1,8-cineole (19.59%), camphor (18.35%), α-pinene (17.17%), camphene (10.10%), β-pinene (6.08%), and α-limonene (3.90%) [35], which is dissimilar in comparison with our findings (40.4%, 11.9%, 8.7%, 3.5%, 6.9%, and 2.4%, respectively).

As mentioned above, EOs are especially known for their variable range of biological functions, including an antioxidant purpose [36]. The antioxidant ability is dependent on compounds that protect the biological system against the deleterious influences of processes causing excessive oxidation exponentiation of reactive oxygen forms [37]. The DPPH, i.e., stable free radical, is a compound often used in methods for determining antioxidants’ free radical scavenging activities [38].

The antioxidant activity of EOs may vary depending on their chemical composition [39]. Our results revealed that the effect of the MEO was stronger as compared to other analysed EOs. This fact may be related to the presence of individual volatile compounds, especially menthol and menthone, containing the hydroxyl radical (-OH), which improves the antioxidant activity strength [40]. We assume based on the data obtained from recent studies that other minor components in MEO, including 1,8-cineole, carvone, and γ-terpinene, could also increase the variable [41,42].

The antimicrobial properties of various plant EOs have been recognized since ancient times [43], and currently, the scientific community is increasingly focused on the evaluation of their capacity to inhibit the growth of diverse foodborne pathogens. Indeed, various studies showed antibacterial properties of many EOs against a wide range of bacterial strains (such as *Stenotrophomonas maltophilia*, *Bacillus subtilis, Y. enterocolitica*, *S. enterica* subs. *enterica*, *Bacillus cereus*, *S. aureus* subs. *aureus*), yeasts (*Candida albicans*, *C. kruseii*, *C. tropicalis*) [20,21,44,45], and fungal strains, including *Penicillium* spp. (*P. citrinum*, *P. crustosum*, *P. expansum*, *P. brevicompactum*, *P. funiculosum*, *P. glabrum*, *P. chrysogenum*, *P. oxalicum*, *P. polonicum*) [46,47].

The antimicrobial activity of *L. officinalis* EO (10 μg/disk) against *P. aeruginosa* was also evaluated in the study by Gavanji et al. [48]. However, the authors found that the zone of inhibition of their EO was 7.83 ± 0.03 mm, and *P. aeruginosa* proved to be more resistant toward a broad range of *L. officinalis* EO concentrations (0.08–100 μg/disk) as compared to *S. aureus*, which is inconsistent with our findings. Indeed, the LEO used in our study possessed a better inhibitory effect on the growth of *P. aeruginosa*, whilst the antibacterial activity against *S. aureus* was weak. This discrepancy between the two studies could be associated with the different chemical compositions of both lavender EOs employed. A particularity of *P. aeruginosa* is its high intrinsic resistance to antiseptics and antibiotics, which is partly caused by its low permeability of the outer membrane [49]. However, the study by Trombetta et al. [50] suggests that the antimicrobial effect of EO components, such as linalool acetate, may (at least partially) result from a perturbation of the lipid fraction of bacterial plasma membranes, thereby leading to alterations of membrane permeability and leakage of intracellular materials. In effect, the amount of linalool acetate was, in our LEO, quantified as a high content, whilst in the EO from *L. officinalis* used in the research by Gavanji et al. [48], it was completely absent. The hypothesis is also supported by the research of Hanamanthagouda et al. [51], in which EOs from dried leaves of *L. bipinnata* containing 3.37% of linalyl acetate exhibited low activity against *P. aeruginosa* (inhibition zone: 7 mm).

REO exhibited the strongest antibacterial activity against *S. aureus* in our research. Gomes Neto et al. [52], in line with this finding, reported significant inhibition of *S. aureus* viability and growth in meat broth induced by the effects of *R. officinalis* EO and by its majority compound 1,8-cineole itself, which was also quantified in our REO in the highest amount. The inhibitory and bactericidal efficiencies of *R. officinalis* EO, with the main component being 1,8-cineole (23.56%), against *S. aureus* were also reported by Jardak et al. [53].

Generally, MIC is a parameter that is often used for the measurement of Eos’ antimicrobial activity, expressing the lowest concentration of the compound able to inhibit the growth of the analysed microorganisms [54].

Our results indicate that even antimicrobial highly resistant isolates, including *S. aureus* [55], *C. albicans* [56], and *E. faecium* [57], showed sensitivity to lavender, mint, and rosemary EOs, predicting their potential usage as promising detergents with the ability to inhibit the growth of a wide range of microorganisms. The differences in the susceptibility of the analysed bacteria and yeasts to the test EOs can be linked to variation in the rate of samples’ penetration through the cell wall and cell membrane structures [58]. In addition to the EOs used, their concentration, and the type of microorganism tested, the differences in MIC values may be influenced by the cell size, cell damage, and EO oxidation [59].

EOs are known for their hydrophobicity through which they are capable of interacting with the fungal plasma membrane, leading to disruption of the membrane structures (leakage of some cellular components) or to alterations of the membrane permeability, reflecting their antifungal effects [60].

Many studies have shown strong antifungal activity of diverse plant EOs with a wide inhibition spectrum, pointing out their high potential as innovative preservative agents to replace synthetic fungicides [61]. Among other factors, the variations in the fungicidal activity of these aromatic compounds can be related to their active compounds, such as phenols, aldehydes, and ketones [62], which is consistent with the GC-MS analysis carried out in our study.

The results from MC and measurements showed that the parameters of bread analysed in our study had values of 41.65 ± 0.55% and 0.944 ± 0.001, respectively. Bread is considered as an intermediate moisture food, with MC typically ranging from 35–42%, and a_w_ above 0.95, which is consistent with our findings. Therefore, baked goods, including bread, are susceptible to microbial spoilage with high growth of various fungi strains [63], thereby offering its application as a suitable type of substrate in such experiments.

Generally, antimicrobial agents are applied in food products for two main reasons: (i) to control natural spoilage processes (preservation of food), and (ii) to prevent or control the growth of microorganisms (food safety) [43]. Our study was focused primarily on an evaluation of the antifungal effects of EOs on bread as a model substrate for fungal growth (in situ conditions) since fungal spoilage occurs more often than bacterial spoilage [64]. Vapour diffusion exposure was applied based on the fact that most of the antimicrobial activity of EOs is attributed to volatile compounds [65].

The EOs’ antifungal activity upon solution contact (broth dilution and agar dilution methods) has been studied by many researchers. However, the activity by vapour-phase contact has been reported more rarely [66,67]. Different types of fungi, including *Penicillium* spp., are responsible for bread spoilage [68]. Although *P. expansum* is mainly associated with the degradation of apples, it was used in our study for its higher resistance than other species of the *Penicillium* strains [69]. Despite the high resistance, the antifungal effectiveness of all EOs tested against *P. expansum*, ranging from 36.05 ± 1.73% (250 µL/L of REO) to 86.48 ± 3.55% (125 µL/L of REO), was reported in the current research. Therefore, we assume that the EOs used can also be effective against other resistant species of microorganisms. Interestingly, *P. citrinum* was the most sensitive to the lowest concentration (125 µL/L) of MEO. We propose that the finding can be associated with the lower concentration of methyl acetate as compared to the MEO higher concentrations used as the chemical compound decreases the antifungal activity of MEO [28]. The results are in accordance with our previous studies, in which the antifungal effects of other EOs, such as coriander EO [20] or *Citrus aurantium* EO [21], against the same fungi species analysed (*P. citrinum*, *P. expansum*, *P. crustosum*) were confirmed.

## 4. Materials and Methods

### 4.1. Essential Oils

The following EOs were applied in this study: Lavender (LEO; *Lavandula angustifolia x latifolia*), mint (MEO; *Mentha x piperita* L.), and rosemary (REO; *Rosmarinus officinalis*). All essential oils were purchased from Hanus s.r.o. (Nitra, Slovakia) to complement our previous results [20,21] from such experiments, thus creating a comprehensive view of the biological actions of various commercially available EOs obtained from the same company.

### 4.2. Fungal Strains

Three *Penicillium* (*P.*) strains (*P. crustosum*, *P. citrinum*, *P. expansum*) were isolated from berry samples of *Vitis vinifera* and consequently classified using a reference-based MALDI-TOF MS Biotyper. The obtained results were also validated by comparison with the taxonomic identification obtained by 16S rDNA sequences analysis.

### 4.3. Microbial Strains

Three Gram-negative bacteria: *P. aeruginosa* CCM 1959, *S. enterica* subsp. *enterica* CCM 3807, *Y. enterocolitica* CCM 5671; two Gram-positive bacteria: *E. faecalis* CCM 4224, *S. aureus* subsp. *aureus* CCM 4223; and four yeasts: *C. glabrata* CCM 8270, *C. albicans* CCM 8186, *C. krusei* CCM 8271, and *C. tropicalis* CCM 8223 were used to evaluate the antimicrobial activities of the EOs. The microorganisms were obtained from the Czech Collection of Microorganisms (Brno, Czech Republic).

### 4.4. Evaluation of Antioxidant Activity of the EOs

The antioxidant activity of the three analysed EOs was assessed on the basis of the scavenging activity of the stable radicals 2,2-diphenyl-1-picrylhydrazyl (DPPH) according to the methodology used in the studies [20,21].

### 4.5. Chemical Characterization of EO Samples by Gas Chromatography/Mass Spectrometry (GC/MS) and Gas Chromatography (GC-FID)

Gas chromatography/mass spectrometry analyses of the selected EO samples were performed using an Agilent 6890N gas chromatograph (Agilent Technologies, Santa Clara, CA, USA) coupled to a quadrupole mass spectrometer 5975B (Agilent Technologies, Santa Clara, CA, USA). A HP-5MS capillary column (30 m × 0.25 mm × 0.25 m) was used. The temperature program was as follows: 60 °C to 150 °C (increasing rate 3 °C/min) and 150 °C to 280 °C (increasing rate 5 °C/min). The total run time was 60 min. Helium 5.0 was used as the carrier gas with a flow rate of 1 mL/min. The injection volume was 1 L (EO samples were diluted in pentane), while the split/splitless injector temperature was set at 280 °C. The investigated samples were injected in the split mode with a split ratio at 40.8:1. Electron-impact mass spectrometric data (EI-MS; 70 eV) were acquired in scan mode over the m/z range 35–550. The mass spectrometry ion source and MS quadrupole temperatures were 230 °C and 150 °C, respectively. Acquisition of data started after a solvent delay time of 3 min. Gas chromatography (GC-FID) analyses were performed on an Agilent 6890N gas chromatograph coupled to an FID detector. Column (HP-5MS) and chromatographic conditions were the same as for GC-MS. The FID detector temperature was set at 300 °C.

The individual volatile constituents of injected EO samples were identified based on their retention indices [69], and a comparison with reference spectra (Wiley and NIST databases). The retention indices were experimentally determined using the standard method [70], which included retention times of *n*-alkanes (C6–C34), injected under the same chromatographic conditions. The percentages of the identified compounds (amounts higher than 0.1%) were derived from their GC peak areas.

### 4.6. Evaluation of Antimicrobial Activity of the EOs

The evaluation of the antimicrobial activity of the EOs was performed using the agar disc diffusion method. For this purpose, there was an aliquot of 0.1 mL of fungal and bacterial suspension in Mueller Hinton Broth (MHB; Merck, Gernsheim, Germany) inoculated to Mueller Hinton Agar (MHA; Merck, Germany; 60 mm). Subsequently, the discs of filter paper (6 mm) were impregnated with 10 µL of the analysed EO samples and then applied on the MHA surface. Inoculated MHA plates were kept at 4 °C for 2 h and incubated aerobically at 37 °C for 24 h (bacteria). In the case of fungi, 10 µL of the analysed oils were applied at three concentrations (125, 250, and 500 µL/L, diluted in 0.1% dimethyl sulfoxide (DMSO)), and incubated at 25 °C for 5 days. Two antibiotics (Cefoxitin, Gentamicin) and one antifungal (Fluconazole) were used as positive controls for Gram-negative and Gram-positive bacteria and yeasts, respectively. Disks impregnated with ethanol served as negative controls.

The diameters of the inhibition zones were measured in mm after incubation. Each test was repeated three times (one repeat reflecting one separate plate).

### 4.7. Determination of Minimum Inhibitory Concentration

The minimum inhibitory concentration (MIC) was detected according to the National Committee for Clinical Laboratory Standards (NCCLS) as it was recently described by Kačániová et al. [20,21]. Chloramphenicol and nystatin, and DMSO served as positive and negative controls, respectively. MIC was detected at 570 nm with a spectrophotometer (Promega Inc., Madison, WI, USA).

### 4.8. Bread Preparation

Wheat bread used for analyses was baked in the Laboratory of Cereal technologies (AgroBioTech Research Centre) according to the methodology described by Kačaniova et al. [20,21].

### 4.9. Moisture Content and Water Activity of Bread

The moisture content (MC) and water activity (a_w_) of bread were measured using the Lab Master a_w_ Standard (Novasina AG; Lachen, Switzerland) and the Kern DBS 60-3 moisture analyzer (Kern and Sohn GmbH, Balingen, Germany), respectively, after the bread cooling [20,21].

### 4.10. In Situ Antifungal Analyses on Bread Model

First, the bread samples were cut into slices with a 150 mm height and placed into 0.5 L sterile glass jars (Bormioli Rocco, Fidenza, Italy). A fungal spore suspension of each strain (in final concentration of 1 × 10^6^ spores/mL) was diluted in 20 mL of sterile phosphate-buffered saline with 0.5% Tween 80 by adjusting the density to 1–1.2 McFarland; 5 μL of inoculum were used for bread inoculation. The EOs in concentrations of 125, 250, and 500 μL/L (EOs + ethyl acetate) were evenly distributed in a volume of 100 µL on a sterile paper–filter disc (6 cm), which was inserted into the cover of the jar, except for the treatment of the control group. The jars were hermetically closed and kept at 25 °C ± 1 °C for 14 days in the dark. The size of the microfungal colonies with visible mycelial growth and visible sporulation [20,21] was evaluated using stereological methods. In this concept, the volume density of the colonies was firstly assessed using ImageJ software (National Institutes of Health, Bethesda, MD, USA), counting the points of the stereological grid hitting the colonies and those falling to the reference space (growth substrate used, i.e., bread). The antifungal activities of the EOs were expressed as the percentage of mycelial growth inhibition (MGI), which was calculated using the formula: MGI = [(C − T)/C] × 100 [71], where C = volume density of the fungal colony in the control group and T = volume density of that in the treatment group.

### 4.11. Statistical Analysis

The data from the analyses was statistically evaluated using Prism 8.0.1 (GraphPad Software, San Diego, CA, USA). One-way analysis of variance (ANOVA) followed by Tukey’s test was used to evaluate the statistical significance of differences between the analysed groups of samples. The level of significance was set at *p* < 0.05. MIC50 and MIC90 values (i.e., concentration causing 50% and 90% reduction of microbial growth) were estimated by the logit analysis.

## 5. Conclusions

The current study evaluated the chemical composition, antioxidant, antibacterial, and antifungal activities of rosemary, lavender, and mint EOs (125, 250, and 500 µL/L concentrations) against selected microorganisms. Our results revealed that MEO possessed the highest DPPH radical-scavenging activity, which was even significantly (*p* < 0.05) different from that of LEO and REO. Considering the antimicrobial activity, the EOs exhibited the strongest inhibitory efficiencies against the growth of *P. aeruginosa* and *C. tropicalis* (LEO), *S. enterica* (MEO), and *S. aureus* (REO). From the fungi strains, MEO (in all concentrations) was able to totally inhibit the growth of *P. expansum* and *P. crustosum*, whilst the growth of *P. citrinum* was completely suppressed only by its lowest concentration. From in situ analysis, REO (125 µL/L) exhibited the lowest MGI of *P. citrinum*, and 500 µL/L of MEO caused the highest MGI of *P. crustosum*. Our results suggest the analysed EOs have promising potential as innovative agents for use in the storage of bakery products to extend their shelf-life. Thus, their combination with other preservatives or modified atmosphere packaging could be a valuable alternative in the food industry. Moreover, our results complement our previous studies, thus creating a comprehensive view of the biological activities of various commercially available EOs obtained from the same company, Hanus s.r.o. (Nitra, Slovakia).

## Figures and Tables

**Figure 1 molecules-26-03859-f001:**
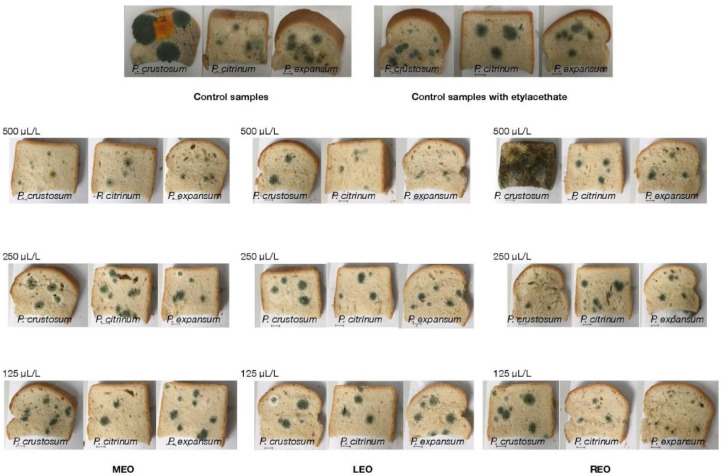
In situ analysis of antifungal activities of the EOs (MEO: Mint essential oil; LEO: Lavender essential oil; REO: Rosemary essential oil).

**Table 1 molecules-26-03859-t001:** Chemical composition of the analysed EOs.

Components	LEO (%)	MEO (%)	REO (%)
1,8-cineole	8.1	5.2	40.4
menthol	-	40.1	-
linalool acetate	35.0	-	-
linalool	32.7	-	1.2
menthone	-	16.8	0.1
camphor	6.4	-	11.9
menthyl acetate	-	9.1	-
α-pinene	0.3	0.7	8.7
β-pinene	0.3	1.1	6.9
neo-menthol	-	4.7	-
(*E*)-caryophyllene	1.6	2.2	5.3
methofuran	-	4.6	-
borneol	2.4	-	3.9
camphene	0.2	tr	3.5
isomenthone	-	2.8	-
α-terpineol	1.3	-	2.7
α-limonene	0.9	1.8	2.4
ocimene	0.3	0.3	2.2
lavandulyl acetate	2.0	-	-
germacrene D	0.4	1.7	tr
pulegone	-	1.5	-
*cis*-sabinene hydrate	-	1.5	0.2
β-myrcene	0.6	0.2	1.5
bornyl acetate	-	-	1.4
4-terpineol	1.3	-	1.1
γ-terpinene	tr	0.5	1.2
(*Z*)-β-farnesene	1.0	-	-
(*E*)-β-ocimene	0.9	tr	0.1
hexyl butanoate	0.9	-	-
geranyl acetate	0.8	-	-
α-humulene	-	-	0.7
3-carvomenthenone	-	0.6	-
α-terpinene	-	0.3	0.6
caryophyllene oxide	-	tr	0.6
sabinene	tr	0.5	0.4
β-bourbonene	-	0.5	-
δ-cadinene	-	0.5	0.3
α-thujene	tr	tr	0.4
α-terpinolene	tr	0.4	0.4
α-copaene	-	tr	0.4
neryl acetate	0.4	-	-
iso-menthyl acetate	-	0.4	-
(*E*)-β-farnesene	-	0.4	-
α-amorphene	0.3	tr	0.2
hexyl tiglate	0.3	-	-
α-bisabolol	0.3	-	-
3-octanol	-	0.3	-
isomenthol	-	0.3	-
bicyclogermacrene	-	0.3	-
trans-linalool oxide	0.3	-	-
3-octanone	0.3	-	-
α-phellandrene	-	-	0.2
δ-3-carene	-	-	0.2
viridiflorol	-	0.2	-
*n*-amyl isovalerate	-	0.2	-
*n*-hexanol	0.1	-	-
pinocarvone	-	-	0.1
tricyclene	tr	-	0.1
*p*-cimene	-	-	0.1
β-elemene	-	tr	-
carvone	-	tr	-
isopulegol	-	tr	-
*cis*-3-hexenol	tr	tr	-
β-thujone	-	-	tr
α-ylangene	-	-	tr
aromadendrene	-	-	tr
3-octanol	tr	-	-
ethyl hexanoate	tr	-	-
*cis*-linalool oxide	tr	-	-
capryl acetate	tr	-	-
nerol	tr	-	-
caryophyllene oxide	tr	-	-
epi-α-cadinol	tr	-	-
**Total**	**99.4**	**99.6**	**99.5**

Note: tr—compounds identified in amounts less than 0.1%; -—not detected.

**Table 2 molecules-26-03859-t002:** Antioxidant activity of the analysed EOs.

	LEO	MEO	REO
Antioxidant Activity (%)	29.08 ± 0.99 ^a^	36.85 ± 0.49 ^b^	28.76 ± 2.68 ^a^

Note: Mean ± standard deviation. MEO: Mint essential oil; LEO: Lavender essential oil; REO: Rosemary essential oil; Values with different superscripts within the same row are significantly different (*p* < 0.05).

**Table 3 molecules-26-03859-t003:** Antimicrobial activity of EOs (inhibition zone in mm).

EOs	Gram-Negative Bacteria			Gram-Positive Bacteria		Yeasts			
	PA	SE	YE	EF	SA	CG	CA	CK	CT
Inhibition Zone [mm]									
LEO	9.3 ± 0.6 ^a^	1.3 ± 0.6 ^a^	1.0 ± 0.0 ^a^	3.0 ± 0.0 ^a^	1.0 ± 0.0 ^a^	2.0 ± 0.0 ^a^	2.0 ± 0.0 ^a^	6.3 ± 0.6 ^a^	9.7 ± 0.6 ^a^
MEO	7.3 ± 1.5 ^b^	9.0 ± 1.0 ^b^	7.0 ± 1.0 ^b^	8.0 ± 1.0 ^b^	5.3 ± 0.6 ^b^	7.3 ± 0.6 ^b^	6.7 ± 0.6 ^b^	9.7 ± 0.6 ^b^	6.0 ± 0.0 ^b^
REO	7.0 ± 1.0 ^b^	5.3 ± 0.6 ^c^	7.7 ± 1.5 ^b^	8.0 ± 1.0 ^b^	10.3 ± 0.6 ^c^	7.7 ± 0.6 ^b^	8.0 ± 1.0 ^b^	10.0 ± 1.0 ^b^	8.3 ± 0.6 ^c^
ATB	22.0 ± 1.0	23.0 ± 1.0	22.0 ± 1.0	25.0 ± 1.0	26.0 ± 1.0	23.0 ± 1.0	24.0 ± 1.0	25.0 ± 1.0	24.0 ± 1.0

Note: Means ± standard deviation. Values followed by superscript within the same column are significantly different (*p* < 0.05). MEO: Mint essential oil; LEO: Lavender essential oil; REO: Rosemary essential oil. *P. aeruginosa*—PA, *S. enterica*—SE, *Y. enterocolitica*—YE, *E. faecium*—EF, *S. aureus*—SA, *C. glabrata*—CG, *C. albicans*—CA, *C. krusei*—CK, *C. tropicalis*—CT. ATB—positive control (Cefoxitin for G^−^, Gentamicin for G^+^, Fluconazole for yeast).

**Table 4 molecules-26-03859-t004:** Antifungal activity of EOs (inhibition zone in mm).

EOs	*P. crustosum*			*P. citrinum*			*P. expansum*		
Conc.	125 (µL/L)	250 (µL/L)	500 (µL/L)	125 (µL/L)	250 (µL/L)	500 (µL/L)	125 (µL/L)	250 (µL/L)	500 (µL/L)
LEO	0.00 ± 0.00 ^aA^	0.00 ± 0.00 ^aA^	2.67 ± 0.58 ^aB^	2.67 ± 0.58 ^aA^	3.33 ± 0.58 ^aAB^	4.00 ± 1.00 ^aB^	2.33 ± 0.58 ^aAB^	2.67 ± 0.58 ^aBC^	4.00 ± 1.00 ^aC^
MEO	N ^bA^	N ^bA^	N ^bA^	N ^bA^	6.67 ± 0.58 ^bB^	9.00 ± 1.00 ^bC^	N ^bA^	N ^bA^	N ^bA^
REO	0.00 ± 0.00 ^aB^	0.00 ± 0.00 ^aB^	0.00 ± 0.00 ^cB^	1.67 ± 0.58 ^aB^	2.33 ± 0.58 ^aB^	4.00 ± 1.00 ^aC^	3.33 ± 0.58 ^aB^	5.67 ± 0.58 ^cC^	6.00 ± 1.00 ^aC^

Note: Means ± standard deviation. MEO: Mint essential oil; LEO: Lavender essential oil; REO: Rosemary essential oil. Values in the same column with different small letters, and those in the same row with different upper-case letters are significantly different (*p* < 0.05). Conc.—concentration; 0.00—total growth; N—without growth.

**Table 5 molecules-26-03859-t005:** Antimicrobial activity of EOs expressed as the minimum inhibitory concentration (MIC) in µL/mL against Gram-negative and Gram-positive bacteria.

EOs	Gram-Negative Bacteria						Gram-Positive Bacteria			
	PA		SE		YE		EF		SA	
	MIC 50 (µL/mL)	MIC90 (µL/mL)	MIC 50 (µL/mL)	MIC90 (µL/mL)	MIC 50 (µL/mL)	MIC90 (µL/mL)	MIC 50 (µL/mL)	MIC90 (µL/mL)	MIC 50 (µL/mL)	MIC90 (µL/mL)
**LEO**	232.15 ± 0.58 ^aA^	288.41 ± 0.23 ^aB^	94.26 ± 0.19 ^aC^	115.15 ± 0.96 ^aD^	243.11 ± 1.01 ^aE^	391.10 ± 0.23 ^aF^	134.18 ± 0.22 ^aG^	255.21 ± 0.98 ^aH^	205.88 ± 0.14 ^aI^	286.99 ± 1.05 ^aB^
**MEO**	2128.30 ± 0.41 ^bA^	598.41 ± 0.74 ^bB^	5.72 ± 0.44 ^bC^	4.12 ± 0.15 ^bD^	297.96 ± 0.17 ^bE^	255.95 ± 0.07 ^bF^	270.68 ± 0.81 ^bG^	513.86 ± 0.69 ^bH^	19.42 ± 0.62 ^bI^	7.33 ± 0.46 ^bJ^
**REO**	134.51 ± 0.19 ^cA^	155.18 ± 0.09 ^cB^	93.58 ± 0.25 ^cC^	98.75 ± 0.11 ^cD^	255.95 ± 0.65 c^E^	299.76 ± 0.35 ^cF^	270.68 ± 0.73 ^bG^	313.86 ± 0.05 ^cH^	198.58 ± 0.66 ^cI^	331.18 ± 0.41 ^cJ^

Note: Means ± standard deviation. MEO: Mint essential oil; LEO: Lavender essential oil; REO: Rosemary essential oil; *P. aeruginosa*—PA, *S. enterica*—SE, *Y. enterocolitica*—YE, *E. faecium*—EF, *S. aureus*—SA. Values in the same column with different small letters, and those in the same row with different upper-case letters are significantly different (*p* < 0.05).

**Table 6 molecules-26-03859-t006:** Antimicrobial activity of EOs expressed as the minimum inhibitory concentration (MIC) in µL/mL against yeasts.

EOs	Yeasts							
	CG		CA		CK		CT	
	MIC 50 (µL/mL)	MIC90 (µL/mL)	MIC 50 (µL/mL)	MIC90 (µL/mL)	MIC 50 (µL/mL)	MIC90 (µL/mL)	MIC 50 (µL/mL)	MIC90 (µL/mL)
**LEO**	179.61 ± 0.23 ^aA^	241.63 ± 0.11 ^aB^	432.40 ± 0.38 ^aC^	724.99 ± 0.77 ^aD^	121.35 ± 0.17 ^aE^	226.40 ± 0.14 ^aF^	144.25 ± 0.49 ^aG^	191.35 ± 0.46 ^aH^
**MEO**	562.30 ± 0.92 ^bA^	944.85 ± 0.55 ^bB^	459.91 ± 0.73 ^bC^	644.58 ± 0.54 ^bD^	5.50 ± 0.12 ^bE^	8.60 ± 0.07 ^bF^	432.40 ± 0.88 ^bG^	139.81 ± 0.32 ^bH^
**REO**	121.86 ± 0.47 ^cA^	151.83 ± 0.67 ^cB^	459.91 ± 0.56 ^bC^	644.51 ± 0.33 ^bD^	120.38 ± 0.64 ^aE^	296.18 ± 0.09 ^cF^	136.58 ± 0.76 ^cG^	185.45 ± 0.82 ^cH^

Note: Means ± standard deviation. *Candida glabrata*—CG, *Candida albicans*—CA, *Candida krusei*—CK, *Candida tropicalis*—CT. Values in the same column with different small letters, and those in the same row with different upper-case letters are significantly different (*p* < 0.05).

**Table 7 molecules-26-03859-t007:** Mycelial growth inhibition of the analysed EOs.

Fungi Strains	MGI (%)								
	LEO (µL/L)			MEO (µL/L)			REO (µL/L)		
	125	250	500	125	250	500	125	250	500
*P. crustosum*	81.18 ± 2.78 ^aA^	85.88 ± 1.95 ^aA^	88.64 ± 2.74 ^aA^	87.91 ± 1.06 ^aA^	77.65 ± 1.32 ^aB^	90.19 ± 2.99 ^aA^	73.73 ± 0.99 ^aA^	92.48 ± 1.69 ^aB^	-4.71 ± 2.18 ^aC^
*P. citrinum*	61.12 ± 2.59 ^bA^	70.49 ± 1.96 ^bB^	89.38 ± 2.05 ^aC^	42.54 ± 3.11 ^bA^	18.30 ± 3.02 ^bB^	23.03 ± 1.01 ^bC^	14.03 ± 3.37 ^bA^	39.02 ± 4.02 ^bB^	57.36 ± 2.63 ^bC^
*P. expansum*	63.45 ± 3.08 ^bA^	77.62 ± 1.33 ^cB^	86.12 ± 3.04 ^aC^	62.68 ± 1.66 ^cA^	67.05 ± 2.84 ^cA^	82.07 ± 1.65 ^cB^	86.48 ± 3.55 ^cA^	36.05 ± 1.73 ^bB^	41.10 ± 1.77 ^cC^

Note: Means ± standard deviation. MEO: Mint essential oil; LEO: Lavender essential oil; REO: Rosemary essential oil. Values in the same column with different small letters, and those in the same row with different upper-case letters are different (*p* < 0.05).

## Data Availability

Data is contained within the article.

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
