# Peer review of "In Vitro Antimicrobial Activity of Lavender, Mint, and Rosemary Essential Oils and the Effect of Their Vapours on Growth of Penicillium spp. in a Bread Model System"

_molecules, 2021, doi:10.3390/molecules26133859_

Round 1
Reviewer 1 Report
Manuscript Number: molecules-1259595
Title: Rosemary, Lavender and Mint Essential Oils: Antimicrobial and the Vapour-phase Antifungal Activities against Selected Microorganisms
The manuscript describes the antioxidant, antimicrobial and antifungal activities of Rosemary, Lavender and Mint essential oils on three Penicillium spp. and several microorganisms selected from gram-negative, gram-positive bacteria and yeasts. The topic of the influence of essential oils on the growth of various microorganisms has been widely investigated recently, nevertheless in my opinion the presented study is a certain extension of the existing base of knowledge. The paper is quite fairly written and organized however some points need to be explained or improved. Below, there are some comments and suggestions:
lines 124-133: It is not necessary to repeat the data from the Table 3 in the text in brackets.
Table 3: One decimal place in the given values is enough, especially since it is the average of the measurements made with a resolution of 1 millimeter.
Subsection 2.4.2. MIC against Microscopic Fungi (lines 172-190): The MIC (Minimal Inhibitory Concentration) is a quantitative indicator, while the MIC for moulds determined in the submitted study is not a fully quantitative indicator, because we do not know how the volume of 10 µl of analyzed oils at 125, 250, and 500 µl/l was translated into on the concentration of essential oils in the target environment. It would be better to drop point “2.4.2. MIC against Microscopic Fungi”, and its content should be attached to point 2.3. Antimicrobial Activity of Eos (after merging both points, the title of the section can be changed to reflect the content). This type of procedure is used in the methodology, where these elements are described together (405-416).
Table 6: How can authors explain the fact that lower concentration of MEO (125 μL/L) inhibited the growth of P. citrinum, while higher concentrations did not? Moreover, there is a lack of EOs concentration units in Table 6.
line 201: Shouldn’t it be “(89.38…)” instead of “(9.38….)”
line 405: In the methodology, it would be advisable to determine the diameter of the agar plates. This is important in the context of mould and Table 6, in which it is reported that in case of MEO for P. crustosum, P. expansum and P. citrinum (125 µl/l) no growth was observed (but what was the plate size, on which no growth was observed). Besides, there was lack how the repetitions were carried out (each of repetitions was on a separate plate, or all repetitions were made at certain distance from each other on the same plate) - an illustrative photo would be useful, as in the case of bread.
line 410: “…10 μL of analyzed EO samples and then inoculated on the MHA surface…”
and lines 412-413: ”…10 μL of analyzed oils were inoculated at three concentrations (125 μL/L, 250 μL/L, and 500 μL;…”
The authors misuse the word "inoculation" for the operation of placing soaked with essential oils filter paper discs on the agar. The word "inoculation" is usually used to the introduction of a microbial suspension to stimulate certain specific state/conditions. The use of the word "inoculation" in mentioned context may be confusing, particularly that microbial inoculation of agar also occurs in the manuscript.
line 437: “…The EOs in concentrations of 125, 250, and 500 μL/L (EOs + ethyl acetate) were evenly distributed on a sterile paper–filter disc (6 cm)…” – information about the volume of used essential oils has not been included in the description.
Author Response
The authors are very grateful to the Reviewer for valuable comments. We would like to thank the Reviewer for the time devoted for constructive and important comments to improve our paper. All changes in the manuscript have been done.
Point 1: lines 124-133: It is not necessary to repeat the data from the Table 3 in the text in brackets.
Response: Edited directly in the manuscript.
Point 2: Table 3: One decimal place in the given values is enough, especially since it is the average of the measurements made with a resolution of 1 millimeter.
Response: The values were changed directly in the Table.
Point 3: Subsection 2.4.2. MIC against Microscopic Fungi (lines 172-190): The MIC (Minimal Inhibitory Concentration) is a quantitative indicator, while the MIC for moulds determined in the submitted study is not a fully quantitative indicator, because we do not know how the volume of 10 µl of analyzed oils at 125, 250, and 500 µl/l was translated into on the concentration of essential oils in the target environment. It would be better to drop point “2.4.2. MIC against Microscopic Fungi”, and its content should be attached to point 2.3. Antimicrobial Activity of Eos (after merging both points, the title of the section can be changed to reflect the content). This type of procedure is used in the methodology, where these elements are described together (405-416).
Response: Edited directly in the manuscript.
Point 4: Table 6: How can authors explain the fact that lower concentration of MEO (125 μL/L) inhibited the growth of P. citrinum, while higher concentrations did not? Moreover, there is a lack of EOs concentration units in Table 6.
Response: Completed directly in the manuscript. Since we follow strictly ethical standards in presentations of our research, we wrote in text of manuscript right experimental values. Of course, here, also MIC values. In many situations researchers observed that higher concentrations of plant extracts or essential oils (even pure compounds) exert opposite effect in comparison to lower concentrations (not only in such situations, e.g., sometimes lower concentrations also induce cells dead, whiles in higher concentrations induced cells proliferation). So, what is point here: Presented MIC values suggests that examined samples, in correctly adjusted concentration, could be useful in treatment of fungal growth and infection.
Point 5: line 201: Shouldn’t it be “(89.38…)” instead of “(9.38….)”
Response: Edited directly in the manuscript.
Point 6: line 405: In the methodology, it would be advisable to determine the diameter of the agar plates. This is important in the context of mould and Table 6, in which it is reported that in case of MEO for P. crustosum, P. expansum and P. citrinum (125 µl/l) no growth was observed (but what was the plate size, on which no growth was observed). Besides, there was lack how the repetitions were carried out (each of repetitions was on a separate plate, or all repetitions were made at certain distance from each other on the same plate) - an illustrative photo would be useful, as in the case of bread.
Response: The data are supplemented in a manuscript. We do not have photos from the experiment, in future experiments we will think about this fact for the sake of clarity of the results.
Point 7: line 410: “…10 μL of analyzed EO samples and then inoculated on the MHA surface…”
and lines 412-413: ”…10 μL of analyzed oils were inoculated at three concentrations (125 μL/L, 250 μL/L, and 500 μL;…”
Response: Edited directly in the manuscript.
Point 8: The authors misuse the word "inoculation" for the operation of placing soaked with essential oils filter paper discs on the agar. The word "inoculation" is usually used to the introduction of a microbial suspension to stimulate certain specific state/conditions. The use of the word "inoculation" in mentioned context may be confusing, particularly that microbial inoculation of agar also occurs in the manuscript.
Response: Edited directly in the manuscript.
Point 9: line 437: “…The EOs in concentrations of 125, 250, and 500 μL/L (EOs + ethyl acetate) were evenly distributed on a sterile paper–filter disc (6 cm)…” – information about the volume of used essential oils has not been included in the description.
Response: Edited directly in the manuscript.
Reviewer 2 Report
The article presents the antioxidant and antimicrobial properties of three commercially available essential oils: rosemary (REO), lavender (LEO) and mint (MEO), the tests performed by the authors being the usual ones, without having a note of originality.
Of course, the objective of finding bread preservatives from natural sources is very important, but I think they should not be sought among these EOs, because their main qualities are taste and smell, which significantly changes the organoleptic properties of bread.
That is why I consider that the research carried out is not justified and the objective was not well chosen. The study may be more interesting if it focused on other EOs.
Also, the chemical composition of these EOs is probably made by the manufacturer, considering that they are commercial products. It would be interesting for the authors to find new plant resources for obtaining EOs with a preservative effect. The antioxidant and antibacterial activity of these EOs have been intensively studied over time, and the results of the tests performed only confirm what was already known.
I appreciate the research carried out, but I consider that is only a preparatory stage of a work that really has importance for the specialists in the field.
Author Response
Reviewer #2
The authors are very grateful to the Reviewer for valuable comments. We would like to thank the Reviewer for the time devoted for constructive and important comments to improve our paper. All changes in the manuscript have been done.
Point 1: The article presents the antioxidant and antimicrobial properties of three commercially available essential oils: rosemary (REO), lavender (LEO) and mint (MEO), the tests performed by the authors being the usual ones, without having a note of originality.
Response: Our research was aimed to investigate antifungal and antibacterial activities of the three EOs (REO, LEO, MEO) in order to create a comprehensive view of such effects of various commercially available EOs (e.g coriander, orange, …) obtained from the same company, Hanus s.r.o. (Nitra, Slovakia).
Point 2: Of course, the objective of finding bread preservatives from natural sources is very important, but I think they should not be sought among these EOs, because their main qualities are taste and smell, which significantly changes the organoleptic properties of bread.
Response: Indeed, EOs used could change the organoleptic properties of bread; however, due to the intake of chemical preservatives, consumer perception towards natural antimicrobial food preservatives is intensively changing. Moreover, researches evaluated the sensory properties of bread enriched with lavender, rosemary and mint powder in adequate levels have revealed their acceptability by consumers (Vasileva et al., 2018; Ali et al., 2019; Shori et al., 2021, respectively).
References:
Vasileva, I., Denkova, R., Chochkov, R., Teneva, D., Denkova, Z., Dessev, T., ... & Slavov, A. (2018). Effect of lavender (Lavandula angustifolia) and melissa (Melissa Officinalis) waste on quality and shelf life of bread. Food chemistry, 253, 13-21.
Altae Abd Ali, A. (2019). Effect of Addition of Rosemary Leaves Powder on the Rheological Characteristics of Dough in Addition to the Quality Attributes of Bread Manufactured from to Local Wheat. Journal of University of Babylon for Pure and Applied Sciences, 27(1), 118-126.
Shori, A. B., Kee, L. A., & Baba, A. S. (2021). Total Phenols, Antioxidant Activity and Sensory Evaluation of Bread Fortified with Spearmint. Arabian Journal for Science and Engineering, 46(6), 5257-5264.
Point 3: That is why I consider that the research carried out is not justified and the objective was not well chosen. The study may be more interesting if it focused on other EOs.
Response: How it was already mentioned, it was necessary to choose also the EOs from lavender, mint and rosemary to create a comprehensive view of such effects of various commercially available EOs obtained from the same company, Hanus s.r.o. (Nitra, Slovakia). Completely results will be finally published in the form of scientific review.
Point 4: Also, the chemical composition of these EOs is probably made by the manufacturer, considering that they are commercial products. It would be interesting for the authors to find new plant resources for obtaining EOs with a preservative effect. The antioxidant and antibacterial activity of these EOs have been intensively studied over time, and the results of the tests performed only confirm what was already known.
Response: We agree with the reviewer. Indeed, to find new plant resources for obtaining EOs with a preservative effect will be our next challenge. Also, such experiments have been in fact already performed but it is important to note that these effects are strongly connected with the chemical composition of the EOs which depends on many serious factors (such as the plant developmental state, the plant part used for extraction, plant geographical location, and physical and chemical characteristics of the soil and climate in question, …) influencing their ultimate antibiological activities. Therefore, discrepancies in the results among the studies are usually occurred.
Point 5: I appreciate the research carried out, but I consider that is only a preparatory stage of a work that really has importance for the specialists in the field.
Response: We hope that our results will be beneficial for scientific community with more experience in this field of the research.
Reviewer 3 Report
Dear Authors,
I have reviewed the manuscript entitled "Rosemary, Lavender and Mint Essential Oils: Antimicrobial and the Vapour-Phase Antifungal Activities Against Selected Microorganisms" and found it to be well written and of interest for the scientific community as in presents the composition, antioxidant and antimicrobial activities of rosemary, lavender and mint essential oils that could find valuable applications in the food industry as preservatives.
There are some small observations to be addressed in order to improve the quality of the manuscript, that are mentioned later:
- Please write "in situ" in italics throughout the entire manuscript (see lines 75, 344, 466).
- Line 66: I recommend changing "therapeutic" (singular) to "therapeutics" (plural).
- Line 77: Please use "was" (past tense) instead of "will" (future tense).
- Line 83: Please correct "Eos" to "EOs".
- Line 102: Please add "%" after "±2.68".
- Line 114: I recommend the following change: "zone of inhibition of 9.33±0.58 mm".
- Line 152: I suggest the following change: "against C. glabrata and C. albicans. On the other hand, MEO was the most effective".
- Please detail all the abbreviations used in the manuscript, the first time they appear in the text.
- Line 198: I recommend eliminating "significantly" from the sentence.
- Line 201: I believe "9.38 ± 2.05%" should be replaced with "89.38 ± 2.05%".
- Line 205: I recommend eliminating "considerably" from the sentence.
- Line 254: I suggest the following change: "other studies [31-33] in which".
- Line 298: Please change "results" to "result".
- Line 314: I suggest the following change: "expressing the lowest concentration of the compound able to inhibit".
- Line 336: I recommend using "food" instead of "foods".
- Line 414: Please change to: "Two antibiotics (cefoxitin and gentamycin" and one antifungal (fluconazole) were used".
- Line 422: Please correct "Chloram-phenicol" to "Chloramphenicol".
- Why didn't you use the same reference antimicrobials for assessing the antimicrobial activities of the EOs? Why didn't you include the values obtained for the positive controls in the tables with the results comprising the antimicrobial properties?
Author Response
Reviewer #3
The authors are very grateful to the Reviewer for valuable comments. We would like to thank the Reviewer for the time devoted for constructive and important comments to improve our paper. All changes in the manuscript have been done.
There are some small observations to be addressed in order to improve the quality of the manuscript, that are mentioned later:
Point 1: Please write "in situ" in italics throughout the entire manuscript (see lines 75, 344, 466).
Response: Edited directly in the manuscript.
Point 2: Line 66: I recommend changing "therapeutic" (singular) to "therapeutics" (plural).
Response: Edited directly in the manuscript.
Point 3: Line 77: Please use "was" (past tense) instead of "will" (future tense).
Response: Edited directly in the manuscript.
Point 4: Line 83: Please correct "Eos" to "EOs".
Response: Edited directly in the manuscript.
Point 5: Line 102: Please add "%" after "±2.68".
Response: Edited directly in the manuscript.
Point 6: Line 114: I recommend the following change: "zone of inhibition of 9.33±0.58 mm".
Response: Edited directly in the manuscript.
Point 7: Line 152: I suggest the following change: "against C. glabrata and C. albicans. On the other hand, MEO was the most effective".
Response: Edited directly in the manuscript?
Point 8: Please detail all the abbreviations used in the manuscript, the first time they appear in the text.
Response: Edited directly in the manuscript.
Point 9: Line 198: I recommend eliminating "significantly" from the sentence.
Response: Edited directly in the manuscript.
Point 10: Line 201: I believe "9.38 ± 2.05%" should be replaced with "89.38 ± 2.05%".
Response: Edited directly in the manuscript.
Point 11: Line 205: I recommend eliminating "considerably" from the sentence.
Response: Edited directly in the manuscript.
Point 12: Line 254: I suggest the following change: "other studies [31-33] in which".
Response: Edited directly in the manuscript.
Point 13: Line 298: Please change "results" to "result".
Response: Edited directly in the manuscript.
Point 14: Line 314: I suggest the following change: "expressing the lowest concentration of the compound able to inhibit".
Response: Edited directly in the manuscript.
Point 15: Line 336: I recommend using "food" instead of "foods".
Response: Edited directly in the manuscript.
Point 16: Line 414: Please change to: "Two antibiotics (cefoxitin and gentamycin" and one antifungal (fluconazole) were used".
Response: Edited directly in the manuscript.
Point 17: Line 422: Please correct "Chloram-phenicol" to "Chloramphenicol".
Response: Edited directly in the manuscript.
Point 18: Why didn't you use the same reference antimicrobials for assessing the antimicrobial activities of the EOs? Why didn't you include the values obtained for the positive controls in the tables with the results comprising the antimicrobial properties?
Response: In our study were two antibiotics for bacteria and one for yeasts used as positive control. The used antimicrobials is depend of bacteria, one was used for Gram-positive and second for Gram-negative bacteria which are listed in EUCAST (EUROPEAN COMMITTEE ON SUSCEBILITY TESTING). Data of positive control were edited directly in the manuscript.
Round 2
Reviewer 2 Report
I believe that the study was performed correctly and is supported by the results obtained, but I still believe that the originality is reduced, as well as the practical applicability of the results.
The researched products are commercial and I think they were intensively studied by the producers before reaching the market.
The research has more of an informative character and is a research model that can be followed.
In appreciation of the effort made by the research team, I agree with the publication.